# Production Forecast of China's New Energy Passenger Vehicles in 2021–2023 under the Compliance of Dual-Credit Policy

**Li Lv and Xi Li \***

China Automotive Technology and Research Center Co., Ltd., Automotive Data of China Co., Ltd., Tianjin 300300, China; lvli@catarc.ac.cn
\* Correspondence: lixi@catarc.ac.cn

**Abstract:** The corporate average fuel consumption (*CAFC*) and new energy vehicle (*NEV*) credit policy (2021–2023) was officially released in June 2020. As a mandatory regulation for automobile manufacturers to produce new energy vehicles, its impact on the output of new energy vehicles needs to be systematically evaluated. In this study, we build an enterprise policy compliance model to simulate the dual-credit policy requirements for the production of new energy vehicles from 2021 to 2023 under different scenarios. The results show that the production of new energy vehicles from 2021 to 2023 is required to reach 1.78 to 3.97 million under different scenarios. Three factors, i.e., switching from New Europe Driving Cycle (NEDC) to World Light Vehicle Test Procedure (WLTP) fuel consumption improvement of conventional vehicles, and credit per new energy vehicle, have a more significant impact on the new energy vehicle production than others. Under the minimum guarantee scenario, a 10% change in the above three factors will lead to a 2.5%, 1.5%, and 0.5% reduction in the production requirement for new energy vehicles.

**Keywords:** China; dual-credit policy; new energy vehicle; production





## 1. Introduction

In order to effectively respond to the energy and environmental issues, promote the development of the new energy vehicle (*NEV*) industry, and accelerate the transformation and upgrading of the Chinese auto industry, in September 2017, five ministries including the Ministry of Industry and Information Technology (MIIT), jointly issued the corporate average fuel consumption (*CAFC*) and new energy vehicle (*NEV*) credit policy (the dual-credit policy) (2017–2020). Phase I of the policy covers the period from 2017 to 2020. With the implementation of the Phase I dual-credit policy coming to an end, in June 2020, the five ministries jointly released the amendments to the 2021–2023 dual-credit policy.

The dual-credit policy carries out the parallel management of *CAFC* and *NEV* credits, and meanwhile generates two kinds of credits, i.e., *CAFC* and *NEV* credits. The assessment targets for the ratio of *CAFC* credits and *NEV* credits are set respectively, and an innovative "linkage" mechanism that connects 2 kinds of credits are proposed to reduce the burden of enterprises and manage the two kinds of credits simultaneously through a unified credit platform, thus achieving two goals as saving energy and reducing consumption of passenger cars and promoting the development of new energy vehicles.

The dual-credit policy has played a positive role in promoting the development of the new energy vehicle industry. According to the data released by the Ministry of Industry and Information Technology of China and China Automobile Industry Association, in terms of the improvement of fuel consumption level, the actual average fuel consumption of traditional energy passenger cars in the industry decreased year by year from 6.87 L/100 km in 2016 to 6.46 L/100 km in 2019, with an average annual decrease of 2.03%. After the new energy passenger cars are included, the actual value of fuel consumption of the industry is reduced from 6.43 L/100 km in 2016 to 5.56 L/100 km in 2019, with an average annual decrease of 4.73%, which is obvious. In terms of increasing the scale of new energy vehicles

production, the total domestic production of new energy passenger cars (including imports) is expected to be about 1.2 million in 2020, with an average annual growth rate of 29% in the past five years; The penetration rate of new energy passenger vehicles is also increasing year by year, from 1.4% in 2016 to 6.1% in 2020 [1,2].

The amendments to the 2021–2023 "*CAFC* and *NEV* Credit Management" put forward the assessment requirements of *NEV* credits for 2021–2023. Some important adjustments have been made to the credit assessment requirements, credit standard, and credit management rules. How can car manufacturers obtain *CAFC* credits and *NEV* credits? What is the compliance strategy? How will the amendments affect the production of new energy passenger cars in the 3 years to come? Which policy indicators and technology indicators will have a greater impact upon the production of new energy passenger cars? This article answers these questions. The research mainly targets 3 aspects: (1) Analyze the credit calculation method and corporate compliance strategy of the *CAFC* and *NEV* Credit Management; (2) Build the impact measurement model of the amendments policy to analyze the compliance status of enterprise credits from 2021 to 2023, and predict the production of new energy vehicles with the implementation of the amendments; (3) Carry out the sensitivity analysis based on the calculation model developed in this article to study the sensitivity of the production of new energy passenger cars to the key indicators of the Amendment, with a view to providing suggestions on further improving the *CAFC* and *NEV* credit management system.

## 2. Literature Review

Studies have proven that the dual-credit policy is effective to compensate for the decline in purchase subsidies and continue promoting the development of the *NEV* industry. Xue Lulu et al. pointed out that without financial subsidies, the new energy vehicles were unable to fully compete with conventional energy vehicles due to higher costs prior to 2025; as a result, the *NEV* market would be affected mainly by policies in the short term [3]. Chen et al. proposed that with the withdrawal of financial subsidies for new energy vehicles, the focus of *NEV* research should be put on the dual-credit policy [4]. Li et al. used a game theoretical model to analyze the impact upon the *NEV* scale of 4 enterprises under the scenarios of implementing the *NEV* financial subsidy policy and Phase I dual-credit policy. They found that compared with financial subsidies, the dual-credit policy could significantly increase the number of new energy vehicles, which was twice as effective as the financial subsidies [5]. Ou et al., after evaluating the impact of the dual-credit policy upon customers' selection and overall profits, found that the dual-credit policy can effectively stimulate the development of pure electric passenger cars in comparison with other policy scenarios [6].

For the method of predicting the output/sales of new energy vehicles, scholars often use the S-shaped growth curve of vehicle ownership [7]. In addition, there are common top-down methods, such as Bu et al., which use Gompertz function to describe the relationship between car ownership and per capita GDP to predict the future car ownership. Further, according to the survival curve of the relationship between vehicle scrap and inventory, the scrap rate is obtained, and finally the vehicle sales volume in the future is calculated [8]. Some scholars also use the dynamic material flow method to establish a prediction model of passenger car scrap volume [9]. In addition, some scholars deduce the number of vehicles to meet the travel demand based on the travel demand, and combined with the law of vehicle scrapping cycle, get the car ownership needed every year in the future, and further get the new increment of vehicles every year [10]. The above methods are common prediction methods for future vehicle ownership or sales, but the impact of industrial policy is not considered in the model. The results are relatively macro and cannot reflect the disturbance of industrial policy. In particular, it has limitations in predicting the special vehicle type of new energy vehicles. Dual-credit policy is an important policy affecting the output of new energy vehicles. Some scholars predict the output scale of new energy vehicles based on the enterprise compliance strategy under the dual-credit policy. Sen et al.

developed an agent-based model to simulate the compliance strategies of the three car manufacturers under different policy scenarios (including CAFE rules and regulations) and estimated the potential market share of electric vehicles in the future [11]. Yu et al. constructed a production competition game model among the Cournot duopoly monopoly enterprises to solve the nash equilibrium, and then analyzed the changes of the optimal production of conventional energy vehicles and new energy vehicles of enterprises with the dual-credit policy [12].

Some scholars have also studied the factors that have a significant impact on the production of new energy vehicles in the dual-credit policy system. Wang et al. constructed a mathematical optimization model that simulates the decision-making process of car enterprises. The study found that super credits, costs of technological upgrade, and the average annual decline rate of fuel consumption of conventional energy vehicles are the most important factors affecting the cost of credit policy compliance [13]. Li et al. established a multi-period dynamic game model to simulate the compliance strategies of three virtual car enterprises. The study shows that enhancing the requirements on the ratio of *NEV* credits will promote the development of new energy vehicles while slowing down the development of conventional energy vehicles [14], but the study failed to use the comprehensive industry data, therefore less significant to guide the industry. Yu et al. found that the *NEV* credit per vehicle, proportion of *NEV* credits, and requirements of annual fuel consumption of enterprises will affect vehicle production [12].

In summary, the current research works on the impact of the dual-credit policy on the development of new energy vehicles from the supply side is still quite limited, and the related research methods are also inadequate. First, due to the limited data available to the public, the research hypotheses of the current game model are somewhat deviated from the actual situation of the industry. Second, though the dual-credit policy attaches great importance to the assessment of the enterprises, the current research works normally takes the whole industry as a research object and fails to take the differences among enterprises into account. Third, the previous research works only analyzed the compliance path or cost of dual-credit policy, with neither evaluation on the *NEV* scale nor the evaluation of the impact of the development of the automobile industry on the implementation effect of the dual-credit policy. Fourth, the previous study mainly focused on the relevant impact of the *CAFC* and *NEV* Credit Management in the first phase (2017–2020), with no assessment on the second phase (2021–2023) involved. In this regard, this article has made some improvements: based on the 2021–2023 Amendment, a policy impact assessment model is established based on the dual-credit policy operation mechanism, and uses the *CAFC* and *NEV* credit management platform to publicize the data of 144 domestic passenger car manufacturers [15]. It mainly analyzes the impact of different decline levels of fuel consumption and development of e-ranges of various electric vehicle models on the production of new energy vehicles after 2020. The prediction results consider the differences of enterprises. The compliance strategy is to simulate enterprises one by one, which is more in line with the actual situation of the industry, has more guidance for the medium and long-term prediction of China's new energy vehicles, and can better predict the risk of policy implementation.

## 3. Method

### 3.1. Formula for Calculating CAFC and NEV Credits

#### 3.1.1. Method to Calculate *CAFC* Credits

The *CAFC* Credit is calculated with the difference between the compliant *FC* and *CAFC* of an enterprise multiplying the production or imports of its passenger cars. When the *FC* is less than the compliant *FC*, *CAFC* credits are generated, and when greater, *CAFC* credit deficits are generated, as shown in Formula (1):

$$C_{CAFC} = (S_{CAFC} - T_{CAFC}) \times TP \tag{1}$$

where $C_{CAFC}$ shows the *CAFC* credit; $S_{CAFC}$ is the compliant *CAFC*, with L/100 km as unit; $T_{CAFC}$ is the *CAFC*, with L/100 km as unit; $TP$ is the passenger car production (excluding export volume) or import volume of enterprise;

The compliant *CAFC* of enterprise is shown in $S_{CAFC}$ Formula (2):

$$S_{CAFC} = \frac{\sum_{i=1}^{N} T_i \times V_i}{\sum_{i=1}^{N} V_i} \times \alpha \tag{2}$$

where $i$ shows the serial number of passenger car vehicle; $T_i$ is the target *FC* to which the $i$ vehicle corresponds (L/100 km); $V_i$ is the production or import volume of the $i$ vehicle; $\alpha$ shows the annual *CAFC* requirements, which is 123%, 120% and 115% respectively from 2021 to 2023.

The *CAFC* of enterprises is shown in $T_{CAFC}$ Formula (3):

$$T_{CAFC} = \frac{\sum_{i=1}^{N} FC_i \times V_i}{\sum_{i=1}^{N} V_i \times W_i} \tag{3}$$

where $FC_i$ is *FC* of the $i$ vehicle; $W_i$ is the weight to the $i$ vehicle, as shown in Table 1.

**Table 1.** Weight of output for vehicles.

| Vehicle Type | 2021 | 2022 | 2023 |
|---|---|---|---|
| New Energy Passenger Cars | 2 | 1.8 | 1.6 |
| Fuel consumption less than 3.2 L/100 km | 1.4 | 1.3 | 1.2 |
| Other passenger cars | 1 | 1 | 1 |

3.1.2. Method to Calculate *NEV* Credits

*NEV* Credit refers to the difference between the *NEV* credits and *NEV* credit compliant target of an enterprise in the current year. If the *NEV* credits of an enterprise are smaller than *NEV* credit compliant target, *NEV* credit deficits will be generated; if more, *NEV* credit surplus generated, as shown in Formula (4):

$$C_{NEV} = C_{NEV-t} - C_{NEV-s} \tag{4}$$

where $C_{NEV}$ is the *NEV* credits of the passenger car enterprises; $C_{NEV-t}$ is the *NEV* credits of an enterprise; $C_{NEV-s}$ is the *NEV* credit compliant target of an enterprise;

The *NEV* credits and *NEV* credit compliant target of enterprises are shown in Formulas (5) and (6):

$$C_{NEV-t} = \sum_{i=1}^{N} C_i \times V_{i-NEV} \tag{5}$$

$$C_{NEV-s} = \beta \times \sum_{i=1}^{N} V_{i-CV} \tag{6}$$

where $C_i$ is *NEV* credits of the $i$ *NEV* vehicle; $V_{i-NEV}$ is annual production and import volume of the $i$ *NEV* vehicles; $V_{i-CV}$ is annual production and import volume of the $i$ conventional energy vehicles; $\beta$ is requirements on *NEV* credit ratio; see Table 2 for details:

**Table 2.** Requirements on *NEV* Credit Ratio 2021–2023.

| Enterprise Type | 2021 | 2022 | 2023 |
|---|---|---|---|
| Production of conventional energy vehicles above 30 k | 14% | 16% | 18% |
| Production of conventional energy vehicles below 30 k | | No requirements | |

3.1.3. Method to Determine the Balance between Credits and Credit Deficits

Balance State of *CAFC* and *NEV* Credits and Credit Deficits in the Industry, as shown in Formula (7):

$$\sum_1^n C_{NEV} + \sum_1^n C_{NEV-p} = \sum_1^n C_{-CAFC} + \sum_1^n C_{-NEV} \tag{7}$$

In the Formula, *n* refers to the number of enterprises; In 2019, there were 144 passenger car makers participating in the accounting; $\sum_1^n C_{NEV}$ refers to the sum of *NEV* credit surplus of enterprises participating in the accounting; $\sum_1^n C_{NEV-p}$ refers to the sum of *NEV* credit surplus of various enterprises over previous years that can be carried over to the accounting year; please refer to *CAFC* and *NEV* Credit Management for details; $\sum_1^n C_{-CAFC}$ refers to the sum of *CAFC* credit deficits that are still required to be offset via the purchase of *NEV* credit surplus, after enterprises use the *CAFC* credit surpluses that are carried over from the previous period and transferred by related enterprises. Please refer to *CAFC* and *NEV* Credit Management for the rules of credit carry-over and transfer by related enterprises; $\sum_1^n C_{-NEV}$ refers to the sum of *NEV* credit deficits of enterprises in the accounting year.

*3.2. Construction of Impact Calculation Model of Dual-Credit Policy*

3.2.1. Model Building Principles

After calculating the fuel consumption points and new energy vehicle points of each enterprise, according to the rules of using different points in the points policy and based on the market share of new energy vehicles of different enterprises, the output of new energy passenger vehicles needed when the industry reaches the balance of positive and negative points is calculated as the minimum requirement of the dual-credit policy for the output of new energy passenger vehicles.

The dual-credit policy has a different impact on the assessment of enterprises that only produce new energy vehicles, only produce traditional fuel vehicles, and produce new energy and fuel vehicles at the same time. Pure new energy enterprises can directly meet the requirements of double integral assessment because there is no production of traditional fuel vehicles, and they are positive integral providers in model calculation. Enterprises with traditional fuel vehicle production need to complete the compliance of fuel consumption and new energy vehicle points. When the scale of new energy production is insufficient and the compliance of the two points is not completed at the same time, they will be the buyer of points. Enterprises that only produce traditional fuel vehicles are all integral purchasers and need to purchase new energy positive integral to achieve compliance. In order to accurately assess the impact of the dual-credit policy, it is necessary to conduct independent accounting and simulate transfer transactions for each enterprise's points.

3.2.2. Assumptions

At present, the assessment of the dual-credit policy includes over 140 passenger car makers. The product structure and the technical level are quite different among them. In order to better evaluate the credits compliance performance of enterprises in between 2021 and 2023, the scenario of enterprises developing the new energy vehicles under the requirements of *CAFC* and *NEV* credits assessment is analyzed. The paper is based on the following assumptions during simulation analysis:

**Assumption 1.** *The changing trend of technical indicators of various enterprises is the same as the overall level of the industry.*

The indicators involved in the prediction of credits accounting of enterprises include the increase in the production of passenger cars, increase in the fuel consumption of conventional energy vehicles due to the switching of WLTC operating conditions, the average annual reduction in the average fuel consumption of conventional energy vehicles, and the value of *NEV* credit per vehicle. Based on the assumption that the development

trend of indicators of various enterprises is consistent with the overall technical indicators of the industry, the *CAFC* and *NEV* credits of enterprises are measured.

**Assumption 2.** *Enterprises with credit deficits use the same credit offset sequence and logic.*

Given the impact of flexible credit measures such as the carry-over and transfer of credits and credit transaction among related enterprises, in order to accurately assess the gap of credit deficits in the industry, it is assumed that the same offset logic is adopted to various enterprises with credit deficits in the following order: first, to offset the credit deficits generated in the current year via the *CAFC* credits generated by the enterprise or carried over from the previous period; if the enterprise fails to be compliant, the credit deficits can be further offset via a series of flexible measures such as the carry-over of *CAFC* credits from the related enterprises; and if the credit deficits are still unable to be fully offset after the flexible credits measures are utilized, *NEV* credits should be purchased to complete the offset.

**Assumption 3.** *Industry information is transparent and smooth, and enterprises will not produce additional new energy vehicles after being compliant.*

It is assumed that credits transfer and transaction prices are transparent among enterprises; when balance is reached between credits and credit deficits, no additional new energy passenger cars are produced.

### 3.2.3. Model Building

Based on the situation of 144 enterprises [15], the paper has measured the compliance situation of credits of various enterprises after 2020 and figured out the production of new energy vehicles required by the industry if the requirements raised in the amendment are met. Based on the actual fuel consumption of each passenger car makers in 2019, the structure of new energy products, the scale of production, etc., the development scenarios of key indicators of various enterprises in between 2021 and 2023 are obtained based on Assumption 1. The future credits situation of each enterprise is evaluated in accordance with the requirements of the amendment and the basis of key indicators of various enterprises. Based on Assumption 2 and Assumption 3, the credit compliance strategy of various enterprises is simulated to calculate the minimum production of new energy vehicles if the assessment requirements of the amendment are satisfied. The policy impact assessment framework and research ideas are shown in Figure 1.

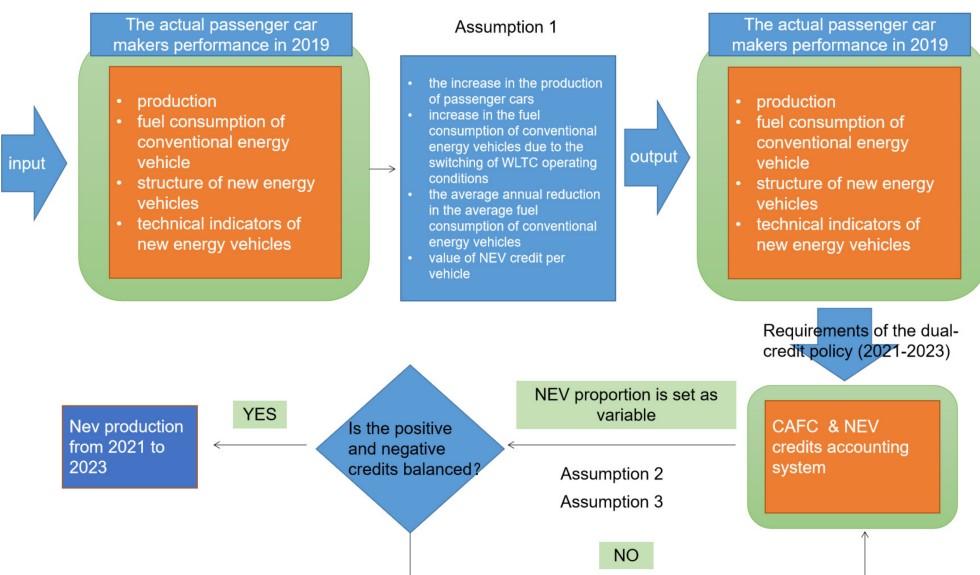

**Figure 1.** Overall assessment framework of impact of the dual-credit policy.

### 3.3. Input Parameters

In the evaluation model analyzed in the paper, the main parameters of *CAFC* and *NEV* credit accounting for passenger car enterprises include the production of passenger cars, the value of *NEV* credits, *CAFC* trend of conventional energy vehicles, etc. Given the large uncertainty in the development of the auto industry, a variety of scenarios are proposed based on some key indicators in order to accurately analyze the effect of the Amendment on the new energy car industry. The parameters involved in the calculation model are shown as below:

### 3.3.1. Total Production of Passenger Cars

Predictions are made based on the laws of the development cycle of the auto industry and the macroeconomic situation. On the basis of passenger car production for 2019 through 2020, the growth rate of passenger car production between 2021 and 2023 is set to be 3.8%, 5%, 5%, respectively, to calculate the production of passenger cars in various years.

### 3.3.2. Credit Value of *NEV* Models

In accordance with the amendment, the benchmark credit value per plug-in hybrid vehicle is 1.6. The credit value of BEVs is related to e-range, power consumption coefficient, etc. In accordance with the Technological Map of Energy Saving and New Energy Vehicles, the development goals of e-range of battery electric passenger cars in 2020 and 2025 are 300 km and 400 km, respectively [11,16]; during the calculation, the 300 km and 400 km are used as two calculation scenarios of average e-range of the battery-electric cars from 2021 to 2023. As to the power consumption of various vehicles, the Development Plan of New Energy Vehicle Industry (2021–2035) officially issued by the State Council proposed that the average power consumption of new energy passenger cars will be reduced to 12.0 kWh/100 km [17] by 2025 and average power consumption coefficient is estimated at about 1.2 times in accordance with the amendment. According to comprehensive calculation, the credit value per battery electric passenger car between 2021 and 2023 can be divided into two scenarios, i.e., 2.6 credits/vehicle and 3.3 credits/vehicle.

### 3.3.3. *FC* Parameters of Conventional Energy Vehicles

Impact of Operating Conditions Switch: After 2021, the test method in the *FC* of conventional energy passenger cars will be switched from NEDC to WLTC, which will increase the approved fuel consumption of conventional energy vehicles on the basis of fuel consumption under NEDC, and based on the fuel consumption standards under Stage V, the target average *FC* of passenger cars will be adjusted from 4 L/100 km to 4.6 L/100 km [18] after 2025, which is up by 15% as a whole. In accordance with the standard compilation instructions under Stage V, the *FC* of conventional energy vehicles will go up by 10.57% after the switch of operating conditions [19]. Based on this, the impact imposed by the switch of the operating conditions in the paper is calculated based on the 3 scenarios of *FC* up by 10%, 12%, and 15% under NEDC.

The decrease in *FC* of Conventional Energy Vehicles. According to the Development Trend of Average Fuel Consumption of Conventional Energy Passenger Cars in China, the average decrease from 2013 to 2018 was 2.08% [15]. If the impact of curb weight and power of various vehicles on the potential of energy saving is not taken into account, the average annual reduction in the *FC* of conventional energy vehicles can reach 2.9% under optimistic conditions [20]. Based on the above factors, when calculating the development trend of fuel consumption of the conventional energy vehicles, the paper sets three scenarios of annual reduction in the *FC* of conventional energy vehicles from 2021 to 2023 as 2%, 2.5%, and 3% respectively.

In summary, when credits are calculated, the *FC* of conventional energy vehicles and credit value of BEV in between 2021 and 2023 are related to different scenarios, as shown in Table 2. Among them, the reduction in the *FC* of conventional energy vehicles has three results of F1–F3, and impact amplitude of switch to WLTC has three results of P1–P3, and

the average credit value of battery-electric vehicles has two results as C1 and C2. Based on the arrangement and combination of the development of aforesaid various indicators, 18 technological development trend scenarios can be figured out, as shown in Table 3.

**Table 3.** Main input parameter scenarios of the policy impact model.

| F: Reduction in *FC* of Conventional Energy Vehicles | P: Impact of Operating Condition Switch | C: Credit Value of Battery Electric Vehicles |
|---|---|---|
| F1: 3% | P1: 10% | C1: 3.2 Credits |
| F2: 2.5% | P2: 12% | C2: 2.5 Credits |
| F3: 2% | P3: 15% | - |

## 4. Results and Discussions

### 4.1. Forecast of Production Requirements of New Energy Passenger Vehicles in 2021–2023

Based on the input parameters of different scenarios in the third part, the accounting results of *CAFC* and *NEV* credits of various enterprises 2021 through 2023 were figured out, and the production of new energy passenger cars when the balance between credits and credit deficits is achieved in the industry is calculated according to the model. The results of various scenarios are shown in Figure 2.

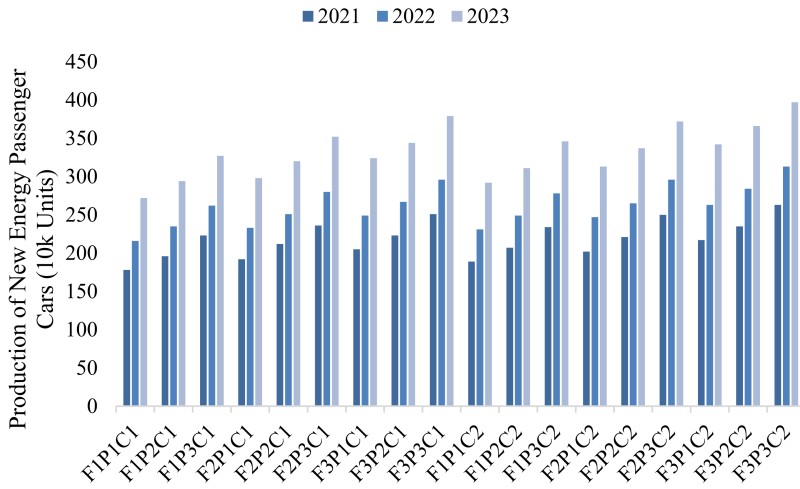

**Figure 2.** Results of Production Demand of New Energy Passenger Cars 2021 through 2023.

There are differences in the production of new energy passenger cars in between 2021 and 2023 under different technological development scenarios, of which the production of new energy passenger cars will be 1780 k to 2630 k in 2021, 2160 k to 3130 k in 2022, and 2720 k to 3970 k in 2023. Taking the reduction of fuel consumption of conventional energy vehicles by 3%, the impact of the switch of operating conditions by 10%, and credit value of BEV, i.e., 3.3 credits (corresponding to F1P1C1 scenario) as the minimum guarantee scenario for the development of the new energy vehicle industry, under this scenario, the production of new energy passenger cars from 2021 to 2023 were 1780 k, 2160 k, and 2720 k respectively, accounting for 8%, 9%, and 11% of the total production of new energy passenger cars respectively.

### 4.2. Relationship between New Energy Passenger Cars Production Requirements and Key Technical Indicators

The research results show that changes in key technical indicators have a direct impact on the production of new energy passenger cars required for credit compliance, as described next.

The impact of operating condition switch is positively correlated with the production of new energy passenger cars. That is, the more significant the effect of the switch is, the

higher the production of new energy passenger cars is required. Taking 2021 as an example, based on the minimum guarantee scenario of a 3% reduction in the fuel consumption of conventional energy vehicles, 10% impact of operating condition switch, and credit value of BEV as 3.3 credits, the production of new energy passenger cars is required to be 1780 k. When the impact of operating conditions switch is 15%, the production of new energy passenger cars will be 2220 k, an increase of 24.7%.

The decrease of fuel consumption of traditional fuel vehicles and the score of pure electric vehicles are negatively correlated with the output of new energy passenger vehicles. The faster the decrease of fuel consumption of traditional fuel vehicles is, the less the output of new energy passenger vehicles is needed. Taking 2021 as an example, the output of new energy passenger vehicles will be 2.05 million, 1.92 million and 1.78 million respectively under the three scenarios of 10% impact of mode switching and 3.3 points of pure electric vehicle, and 2%, 2.5% and 3% reduction of fuel consumption of traditional fuel vehicles.

There is a negative correlation between the score of new energy vehicles and the output of new energy passenger vehicles. The higher the score of new energy vehicles, the less the output of new energy passenger vehicles. Based on the 10% impact of mode switching and the 3% reduction of fuel consumption of traditional fuel vehicles, the production of new energy passenger vehicles in 2021 will be 1.78 million and 1.89 million respectively under the two scenarios of 3.3 points and 2.6 points for pure electric vehicles.

### 4.3. Sensitivity Analysis of Production Requirements of New Energy Passenger Cars to Changes of Key Technical Indicators

In accordance with the pre-set annual reduction in the average fuel consumption of conventional energy vehicles by 2–3%, the impact of operating condition switch by 10–15%, and credit values of BEV as 2.6–3.3 credits, the amplitude of changes in the production demand of new energy passenger cars with the changes of different indicators are calculated, and the differences in the sensitivity of impact of production of new energy passenger cars upon the different indicators are found. The changes in the three indicators are correlated with the changes in the production of new energy passenger cars. Among them, the sensitivity is weakened in turn in accordance with the impact of working conditions impact, annual decrease in fuel consumption of conventional energy vehicles, and credit value of BEV. Based on the minimum guarantee scenario of development of the new energy vehicle industry, if the impact of operating conditions switch is increased by 10%, the production of new energy vehicles will be enhanced by 2.5%; if the average annual reduction in fuel consumption of conventional energy vehicles goes down by 10%, the production of new energy vehicles will be enhanced by 1.5%; and if the average credit value of BEV is decreased by 10%, the production of new energy vehicles will increase by 0.5%, as shown in Figure 3.

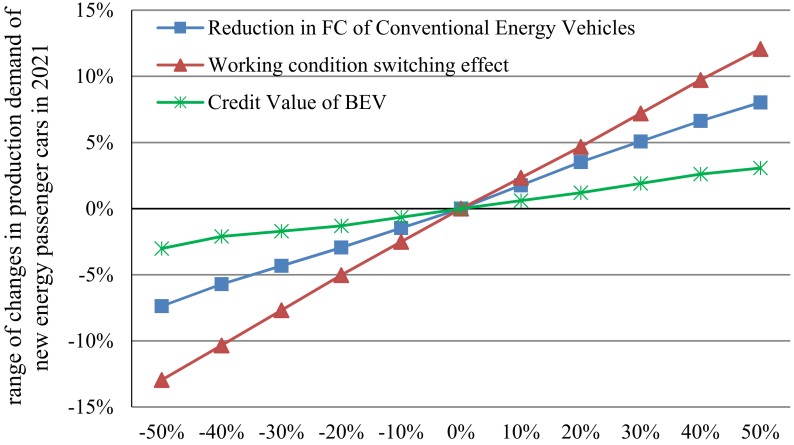

**Figure 3.** Sensitivity Analysis of Key Indicators and Production Demand for New Energy Passenger Cars in 2021 (Minimum Guarantee Scenario).

## 5. Conclusions and Policy Implications

### 5.1. Conclusions

As an institutional innovation of China's automobile industry, the dual-credit policy plays an important role in promoting the rapid development of China's new energy automobile industry. Based on the management requirements of the dual-credit policy amendment and combined with the actual situation of the current industry, this paper establishes a dual-credit policy compliance production prediction model for passenger car enterprises and evaluates the production of new energy passenger cars from 2021 to 2023 under the dual-credit policy amendment by scenario and evaluates the implementation effect of the policy. The main research conclusions and contributions are as follows:

First, this paper establishes a dual-credit policy compliance output prediction model for passenger car enterprises, simulates the compliance strategy of passenger car enterprises under the implementation of dual-credit policy, forecasts the new energy passenger car output under the policy compliance requirements, and evaluates the new energy passenger car output under different technology development scenarios, It provides support for industry development and policy implementation risk prediction.

Second, the results of a simulation study based on the actual data in the paper show that the amendment plays a certain supporting role in the development of the Chinese new energy vehicle industry on the supply side. Under different technology development scenarios, the production of new energy passenger cars between 2021 and 2023 is required to reach 1780–2630 k, 2160–3130 k, and 2720–3970 k, respectively. Under the minimum guarantee scenario of development of the new energy vehicle industry, the production of new energy passenger cars 2021 through 2023 is required to reach 1780 k, 2160 k, and 2720 k, respectively, accounting for 8%, 9%, and 11% of the total *NEV* production respectively.

Third, the paper has found that the key technical indicators as a reduction in fuel consumption of conventional energy vehicles, the impact of operating conditions switch, and credit value of BEV have a more significant impact upon the production of new energy passenger cars. Among them, the range of impact of operating conditions switch is positively correlated with the production demand of new energy vehicles, while the reduction in fuel consumption of conventional energy vehicles and credit value of BEV is negatively correlated with the production demand. The sensitivity of the output of new energy vehicles to the change of fuel consumption during operating condition switching, the decrease of fuel consumption of conventional energy vehicles year by year, and the credit value of BEV are weakened in turn. For every change of 10% in the aforesaid three indicators, the production demand for new energy passenger cars will be affected by 2.5%, 1.5%, and 0.5%, respectively.

In addition to the above research contributions, the research results of this paper also have some limitations and can be improved.

First, this paper has made the assumption that after the number of new energy passenger vehicles produced by the enterprise meets the compliance requirements, it will not produce other new energy passenger vehicles. Therefore, this work only represents the most conservative new energy output under the condition of policy compliance, not the output predicted according to market development. Readers want to know that the output prediction of new energy passenger vehicles should be combined with other research results.

Second, the "enterprises with credit defaults using the same credit Offset sequence and logic" described in assumption 2 is a more optimistic situation. This paper assumes that enterprises carry out negative integral compensation in the order of carry forward, transfer and transaction. In fact, many affiliated enterprises do not necessarily transfer their surplus positive fuel consumption points to affiliated enterprises with negative fuel consumption points but choose to carry forward to the next year for retention. Therefore, in this case, the demand for positive integration of new energy will increase, and the required output of new energy passenger vehicles should also increase.

*5.2. Policy Implications*

The results show that the development level of oil consumption and the growth of new energy production can meet the expectations are the necessary conditions to ensure the smooth implementation of the policy.

The influence of mode switching, and the decrease of fuel consumption of traditional fuel vehicles are the main factors affecting the output requirements of new energy passenger vehicles. Especially for the fuel consumption of traditional fuel vehicles, the annual average decline in history is about 2.03%. According to the 3% minimum scenario, the output of new energy passenger vehicles will also need to reach 1.78 million in 2021. However, there is still some uncertainty whether the target of fuel consumption reduction can be achieved. If the fuel consumption reduction does not meet the expectation, it will further increase the industry compliance requirements for the production of new energy passenger vehicles.

From the historical development of new energy vehicles in China, with the increase of new energy vehicle production base, the growth rate of new energy vehicle production continues to decline, with the growth rate reaching 56–78% from 2016 to 2018, and falling to 3.7% in 2019 [1,2]. According to the minimum scenario, the output of new energy passenger vehicles in 2021 is required to increase by 48.3% compared with the market sales in 2020. Affected by the epidemic situation and declining subsidies [21], there is great uncertainty about whether the scale of output can achieve high-speed growth, which may lead to the risk of insufficient production of new energy vehicles and insufficient supply of new energy positive points.

In view of the above discussion, this paper puts forward two suggestions:

First, it is suggested that the government should not only promote enterprises to increase the production of new energy passenger cars from the supply side through the dual-credit policy, but also stimulate the sales at the consumer side, so as to guarantee the sales at the sales end. We should create a fair consumption environment for the new energy vehicle market, provide tax incentives such as vehicle purchase tax and new energy license right of way incentives, and build perfect charging infrastructure services, so as to reduce the purchase cost and use cost of consumers and accelerate the market development of new energy vehicles.

Second, it is found that the fuel consumption rise caused by switching WLTC condition, the fuel consumption decrease of traditional fuel vehicles and the score of pure electric vehicle have a significant impact on the output prediction of new energy passenger vehicles, and the sensitivity of the output of new energy passenger vehicles to the changes of the three indicators decreases in turn. Therefore, in addition to taking measures to ensure the scale of new energy passenger vehicles, we also need to pay attention to the work of reducing fuel consumption of traditional vehicles. For enterprises, it is necessary to increase the R & D investment in energy-saving technologies such as engine miniaturization and electrification, to reduce the fuel consumption of single vehicle. It is also necessary to adapt to the new test standards and reduce the impact of WLTC condition switching; For the government, it is necessary to strengthen the guidance for the improvement of fuel consumption of traditional fuel vehicles, introduce supporting policies to encourage the development of small vehicles and low fuel consumption vehicles, speed up the decline of fuel consumption of traditional vehicles, and ease the pressure of double points compliance of the industry.

**Author Contributions:** Conceptualization, L.L. and X.L.; methodology, L.L.; software, X.L.; validation, X.L., L.L.; formal analysis, X.L.; investigation, X.L.; resources, X.L.; data curation, L.L.; writing—original draft preparation, X.L.; writing—review and editing, X.L.; visualization, X.L.; supervision, L.L.; project administration, X.L. Both authors have read and agreed to the published version of the manuscript.

**Funding:** This research received no external funding.

**Institutional Review Board Statement:** Not applicable.

**Informed Consent Statement:** Not applicable.

**Data Availability Statement:** Data are contained within this article; more detailed data can be obtained from the corresponding author.

**Acknowledgments:** Thanks to my daughter Shuqiao Wang for her spiritual support of my work and scientific research.

**Conflicts of Interest:** Both authors are employees of China Automotive Technology and Research Center Co., Ltd., Automotive Data of China Co., Ltd. The paper reflects the views of the scientists, and not the company.

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
