# Peer review of "Production Forecast of China’s New Energy Passenger Vehicles in 2021–2023 under the Compliance of Dual-Credit Policy"

_wevj, doi:10.3390/wevj12030119_

Round 1

Reviewer 1 Report

  1. Abstract: It should be about 150-250 words with concise text in a single paragraph. Answer the questions: What problem did you study, and why is it important? What methods did you use? What were your main results? And what conclusions can you draw from your results? Please make your abstract with more specific and quantitative results while it suits broader audiences.
  2. What the paper might lack is a clearer statement of how the model differs from previous works and thus contributes to the literature. Which are the main contributions and the novel characteristics of the proposed methodology to the scientific community? Is there a novel methodological framework, or at least some enhancements to an existing one? The authors should clarify those aspects.
  3. To whom (market participants, systems operators, policymakers) might the work be useful and why? Please provide some examples.
  4. Are there any potential future challenges and/or methodology limitations required to be addressed? All these should be mentioned at the end of the Conclusions section.
  5. Is the provided framework a generic one? Could it be utilized in other case studies without significant modifications?
  6. The authors need to add a flowchart describing the whole investigation procedure to help the readers perceive the main points. In the flowchart, they are recommended to describe the methods used in this study step by step and also link the methods to the corresponding results.
  7. Some additional relevant works might be helpful for the enhancement of the conducted literature review:
    • Bu C, Cui X, Li R, Li J, Zhang Y, Wang C, et al. Achieving net-zero emissions in China’s passenger transport sector through regionally tailored mitigation strategies. Applied Energy. 2021;284:116265.
    • Koltsaklis NE, Dagoumas AS. State-of-the-art generation expansion planning: A review. Applied Energy. 2018;230:563-89.
    • Liu M, Chen X, Zhang M, Lv X, Wang H, Chen Z, et al. End-of-life passenger vehicles recycling decision system in China based on dynamic material flow analysis and life cycle assessment. Waste Management. 2020;117:81-92.

Reviewer 2 Report

Overall the paper is well written and can be considered for publication after a few revisions.

1) Please avoisd abbreviations in the abstract.

2) What are new energy vehicles? Which concepts besides electric vehicles are covered?

3) Assumption 3 is too simplistic as it does not take into account company decisions to take into account future regulatory changes, which definitely will occur. This is the weakest point of the paper and should be either corrected, or an explanation given under which framing conditions this could give insights for the near future.

Round 2

Reviewer 1 Report

The authors have satisfactorily responded to all my questions and made the necessary changes to the manuscript. Their paper can be accepted for publication.